# Rab11A Functions as a Negative Regulator of Osteoclastogenesis through Dictating Lysosome-Induced Proteolysis of c-fms and RANK Surface Receptors

**DOI:** 10.3390/cells9112384

**Published:** 2020-10-31

**Authors:** Yuka Okusha, Manh Tien Tran, Mami Itagaki, Chiharu Sogawa, Takanori Eguchi, Tatsuo Okui, Tomoko Kadowaki, Eiko Sakai, Takayuki Tsukuba, Kuniaki Okamoto

**Affiliations:** 1Department of Dental Pharmacology, Graduate School of Medicine, Dentistry and Pharmaceutical Sciences, Okayama University, Okayama 700-8525, Japan; yokusha@bidmc.harvard.edu (Y.O.); trantienmanh1508@gmail.com (M.T.T.); mami1515@s.okayama-u.ac.jp (M.I.); caoki@md.okayama-u.ac.jp (C.S.); eguchi.takanori@gmail.com (T.E.); 2Department of Radiation Oncology, Beth Israel Deaconess Medical Center, Harvard Medical School, Boston, MA 02115, USA; 3Dental School, Okayama University, Okayama 700-8525, Japan; 4Advanced Research Center for Oral and Craniofacial Sciences, Graduate School of Medicine, Dentistry and Pharmaceutical Sciences, Okayama University, Okayama 700-8525, Japan; 5Department of Oral and Maxillofacial Surgery and Biopathology, Graduate School of Medicine, Dentistry and Pharmaceutical Sciences, Okayama University, Okayama 700-8525, Japan; pphz1rke@okayama-u.ac.jp; 6Department of Frontier Life Science, Graduate School of Biomedical Sciences, Nagasaki University, Nagasaki 815-8582, Japan; tomokok@nagasaki-u.ac.jp; 7Department of Dental Pharmacology, Graduate School of Biomedical Sciences, Nagasaki University, Nagasaki 815-8582, Japan; eiko-s@nagasaki-u.ac.jp (E.S.); tsuta@nagasaki-u.ac.jp (T.T.)

**Keywords:** Rab11A, c-fms, RANK, NFATc-1, osteoclast, vesicular transport

## Abstract

Osteoclast differentiation and activity are controlled by two essential cytokines, macrophage colony-stimulating factor (M-CSF) and the receptor activator of nuclear factor-κB ligand (RANKL). Rab11A GTPase, belonging to Rab11 subfamily representing the largest branch of Ras superfamily of small GTPases, has been identified as one of the crucial regulators of cell surface receptor recycling. Nevertheless, the regulatory role of Rab11A in osteoclast differentiation has been completely unknown. In this study, we found that Rab11A was strongly upregulated at a late stage of osteoclast differentiation derived from bone marrow-derived macrophages (BMMs) or RAW-D murine osteoclast precursor cells. Rab11A silencing promoted osteoclast formation and significantly increased the surface levels of c-fms and receptor activator of nuclear factor-κB (RANK) while its overexpression attenuated osteoclast formation and the surface levels of c-fms and RANK. Using immunocytochemical staining for tracking Rab11A vesicular localization, we observed that Rab11A was localized in early and late endosomes, but not lysosomes. Intriguingly, Rab11A overexpression caused the enhancement of fluorescent intensity and size-based enlargement of early endosomes. Besides, Rab11A overexpression promoted lysosomal activity via elevating the endogenous levels of a specific lysosomal protein, LAMP1, and two key lysosomal enzymes, cathepsins B and D in osteoclasts. More importantly, inhibition of the lysosomal activity by chloroquine, we found that the endogenous levels of c-fms and RANK proteins were enhanced in osteoclasts. From these observations, we suggest a novel function of Rab11A as a negative regulator of osteoclastogenesis mainly through (i) abolishing the surface abundance of c-fms and RANK receptors, and (ii) upregulating lysosomal activity, subsequently augmenting the degradation of c-fms and RANK receptors, probably via the axis of early endosomes–late endosomes–lysosomes in osteoclasts.

## 1. Introduction

Osteoclasts, the bone-resorbing multinucleated cells, were primarily proliferated and differentiated from mononuclear/macrophagic progenitors [1,2,3] via the mechanistic upregulation of c-fms and RANK receptors by binding to M-CSF ligand and RANKL, respectively. RANKL binding to RANK receptor activates nuclear factor of activated T cell cytoplasmic-1 (NFATc-1) [4], which is subsequently translocated into nuclei to promote transcriptionally specific genes prerequisite for bone resorption such as tartrate-resistant acid phosphatase (TRAP), Cathepsin K, and matrix metalloproteinase9 (MMP9) [5,6,7].

The well-characterized superfamily of Ras GTPase proteins is functionally categorized into five cardinal branches such as Ras, Rho/Rac, Arf, Ran, and Rab families. Among these, Rab family of proteins including more than 60 members in human genome regulates a variety of critical steps of membrane trafficking, comprising vesicle movement and transport along actin or tubulin networks, and membrane fusion [8,9,10,11]. Among the Rab GTPases, Rab11 ubiquitously found in eukaryotic cells has been known as a regulator of vesicular trafficking amongst subcellular vesicles, specifically via recycling pathways [12,13,14]. In mammals, the Rab11 subfamily is structurally and functionally composed of three isotypes Rab11A, Rab11B, and Rab11C, the latest of which is also known as Rab25 [15]. Of these, Rab11A GTPase is ubiquitously expressed [16] whereas Rab11B and Rab25 are found exclusively in brain, testis, heart [17], lung, kidney, and gastric tract [18].

Rab7, a small GTPase, enriched in late endosomes and predominantly localized in the perimeter of the ruffled border, functionally served as a suppressor of osteoclast polarization, resulting in abolishment of bone-resorbing activity [19]. Notably, one of our previous studies elucidated an important role of Rab27A in directing delivery of lysosome-mediated organelles and surface receptors modulating osteoclast multinucleation, causing the morphological change to osteoclasts [20]. Later, we also revealed a novel role of Rab44 as a negative regulator of osteoclast differentiation primarily via elevating intracellular Ca^2+^ level, inducing NFATc-1 activation [21]. To Rab11, several previous reports have clarified Rab11A was spatiotemporally concentrated in the pericentriolar endosomal recycling compartments (ERCs) where Rab11 regulated the transport of the recycling transferrin receptor (TfR) [12], whereas Rab11B is localized apically in proximity to the pericentrosomal region distinct from that of Rab11A in MDCK cells [12], suggesting that Rab11A plays a housekeeping role in regulating intracellular pathways of protein transport and/or receptor recycling. Nonetheless, whether Rab11A functionally contributes to cargo transport networks amongst cellular vesicles required for regulation of osteoclast differentiation remains unclear.

As abovementioned, lysosomal functions are required for regulation of osteoclast-induced bone resorption through accelerating the secretion of the specific enzymes such as Cathepsin K, TRAP, and MMP9, essential for bone destruction. In this study, using chloroquine (CLQ), dynamically diffusing into lysosomes and undergoing protonation, causing alkalinization of the lysosomal lumen, thereby impairing lysosomal function [22], we did clarify a novel role of lysosomes in proteolytically degrading c-fms and RANK proteins. Markedly, Rab11A overexpression accelerated the process of lysosomal proteolysis of c-fms and RANK proteins in osteoclasts. Altogether, we speculate one potentiality that Rab11A overexpression promotes lysosome-induced proteolysis of c-fms and RANK receptors mainly through the axis of early endosomes–late endosomes–lysosomes in osteoclasts, causally triggering declination of osteoclastogenesis, eventually stabilizing bone resorption phase.

## 2. Materials and Methods

### 2.1. Antibodies and Reagents

Recombinant human soluble RANKL was isolated by the protocols as described previously [23]. M-CSF was purchased from Kyowa Hakko Kogyo (Tokyo, Japan). Rabbit polyclonal anti-cathepsin B and D antibodies were purified as detailed previously [24]. The other antibodies were used in this study as follows: rat monoclonal anti-LAMP1, (Cat. no. 553,792, BD Biosciences, NJ, USA), rabbit polyclonal anti-c-fms (Cat. no. sc-692, Santa Cruz, CA, USA), monoclonal anti-c-fos (Cat. no. sc-166,940, Santa Cruz, CA, USA), mouse monoclonal anti-NFATc-1 (Cat. no. sc-7294, Santa Cruz, CA, USA), mouse monoclonal anti-RANK (NBP2-247-2, Novus Biologicals Europe, Abingdon, UK), rabbit polyclonal anti-GFP (Green Fluorescent Protein) pAb (Medical and Biological Laboratories Co., LTD., Nagoya, Japan), rabbit polyclonal anti-Rab11A (#2413, Cell Signaling, Danvers, MA, USA), rabbit monoclonal anti-Rab5 (# 3547, Cell Signaling, Danvers, MA, USA) and anti-Rab7 (#9367, Cell Signaling, Danvers, MA, USA), rabbit monoclonal anti-GAPDH (Cat. no. 2118S, Cell Signaling, Danvers, MA, USA).

### 2.2. Cell Culture

A murine monocytic (RAW-D) cell line that was obtained by Prof. Toshio Kukita (Kyushu University, Japan) [25,26] was cultured in minimum essential mediumα (MEMα) (Wako Pure Chemicals, Osaka, Japan) supplemented with 10% fetal bovine serum (FBS), penicillin (100 U/mL), and streptomycin (100 mg/mL). RAW-D cell-derived osteoclasts were induced by RANKL (100–300 ng/mL). Bone marrow-derived macrophages (BMMs) were isolated from the femurs and tibias of 5-week-old male C57BL/6J mice purchased from SLC (Shimizu Laboratory, Japan) as described previously [27], and cultured in MEMα containing M-CSF (50 ng/mL) at 37 °C in 5% CO_2_ overnight. All animal experiments were performed according to the guidelines for the care and use of laboratory animals approved by Okayama University and the Japanese Pharmacological Society (OKU-2018438). Floating cells were collected and cultured on the new culture dishes with MEMα containing M-CSF (50 ng/mL). On day 3 of culture, the adherent cells were referred to as BMMs. The BMMs were refreshed with MEMα supplemented with M-CSF (30 ng/mL) and RANKL (300 ng/mL), and subsequently cultured in the designated time periods.

### 2.3. Western Blot Analysis (WB)

RAW-D cells were pretreated with RANKL over an indicated time course. Cell lysates were obtained using RIPA buffer (50 mM Tris-HCl (pH 8.0), 1% Nonidet P-40, 0.5% sodium deoxycholate, 0.1% SDS, 150 mM NaCl, 1 mM PMSF) including proteinase inhibitor cocktail (Sigma-Aldrich Tokyo, Japan). The protein concentrations were subsequently determined by BCA assay, according to the manufacturer’s guidance (Thermo Pierce, Rockford, IL, USA). The cell lysates (15 µg) were run on 10% SDS-PAGE electrophoresis gels. The proteins were then transferred to PVDF membranes. The blots were blocked in Tris-buffered saline containing 0.05% Tween 20 and 3% skim milk for 1.5 h at room temperature (RT), subsequently probed with various antibodies (1/1000) at 4 °C overnight. After washing membranes, the blots were washed and incubated with horseradish peroxidase (HRP)-conjugate secondary antibodies (GE Healthcare). Blots were eventually detected with ECL substrate (Millipore, Burlington, MA, USA). The immunoreactive bands were observed by the ChemiDoc MP Imaging System (Bio-Rad, Hercules, CA, USA). The quantitative densitometric analysis was performed using Image J.

### 2.4. Small Interfering RNA (siRNA)

The target sequences of murine control the nontargeting siRNA (Ctrl si) (Stealth RNAi^TM^siRNA Negative Control, Invitrogen, Carlsbad, CA, USA) and Rab11A-targeting siRNA (Rab11A si) (Invitrogen Custom Primers, Invitrogen, Carlsbad, CA, USA) were GAAUUCAACCUAGAGAGCAAGAGUA and GACAUCUGCUCUAGAUUCUACAAAU, respectively. RAW-D cells (2 × 10^5^ cells) were seeded and grown in 35 mm culture dishes. The next day, 10 pmol of siRNA was used to transfect into RAW-D cells and was done using Lipofectamine RNAiMAX™ transfection reagent (Invitrogen, Carlsbad, CA, USA), according to the manufacturer’s instructions. On the 1st day of post-transfection, the cell dishes were refreshed by MEMα media supplemented with RANKL. For Western blotting and TRAP staining, the cells were harvested after 3 days of RANKL stimulation.

### 2.5. TRAP Staining

Cells were fixed with 4% paraformaldehyde (PFA) at RT for 60 min and incubated with 0.2% Triton X-100 in PBS at RT for 5 min. Cells were then stained with TRAP solution (0.01% naphtol AS-MX phosphate disodium salt (Sigma-Aldrich, St. Louis, MO, USA), 0.06% fast red violet LB salt (Sigma-Aldrich Tokyo, Japan), 50mM sodium tartrate, and 50mM sodium acetate (pH 5.0)). TRAP-positive counted cells with three or over 10 nuclei in whole well were counted as mature osteoclasts.

### 2.6. Bone Resorption Assay

The bone-resorbing activity of osteoclasts was examined using the Osteo Assay Stripwell Plate (Corning, MA, USA) after stimulated with RANKL for 7 days. Osteoclasts were differentiated from RAW-D cells or BMM cells by RANKL (500 ng/mL) stimulation. Images for bone resorption area and nuclei counting were taken by using all-in-one type fluorescence microscope BZ-9000 (Keyence, Osaka, Japan). Bone resorption area was measured using the Image J software.

### 2.7. Immunocytochemistry

Cells were seeded and grown on glass coverslips and fixed with 4.0% PFA in PBS for 1 h at RT. After washing the cells by PBS, the fixed cells were permeabilized with 0.2% Tween-20 in PBS for 5 min. The cells were incubated with 10% normal goat serum for 30 min and subsequently incubated overnight at 4 °C with the primary antibodies. The cells were washed and stained with the secondary antibodies such as Alexa Fluor 594 goat anti-rat IgG or Alexa Fluor 594 goat anti-rabbit IgG (Cell Signaling Technology Danvers, MA, USA). Finally, nuclear staining with 4,6-diamidino-2-phenylindole (DAPI, Invitrogen Carlsbad, CA, USA) was examined. The samples were visualized using a laser-scanning confocal imaging system (LSM 780 META; Carl Zeiss, AG, Jena, Germany).

### 2.8. Retrovirus Construction and Expression of Mouse Rab11A

The experimental protocols of Retrovirus construction and Rab11A overexpression were performed as described previously [21]. Briefly, the full-length cDNA of mouse Rab11A was generated by PCR using cDNA derived from M-CSF and RANKL-stimulated BMMs for 72 h. The primers were used for GFP forward: 5′-GGACGAGCTGTACAAGGGCACCCGCGACGACGAGTAC-3′, and reverse: 5′-CTACCCGGTAGAATTCTTAGATGTTCTGACAGCACTGC-3′; the cDNAs were amplified by using PrimeSTAR GXL DNA polymerase (Takara, Tokyo) with 40 cycles at which denaturation at 94 °C for 10 s, annealing at 62 °C for 30 s, and extension at 72 °C for 3 min for each cycle. To generate GFP-Rab11A fusion protein, the amplified fragments were fused with a linearized pMSCVpuro-GFP, which was kindly provided by Prof. Kosei Ito (Nagasaki University, Nagasaki, Japan), using In-Fusion cloning kit (Clontech, Mountain View, CA, USA). pMSCVpuro-GFP was also used as a control vector. GFP-alone and/orGFP-Rab11A vector was transfected into HEK293T cells by employing the Lipofectamine 2000 (Life Technologies, Gaithersburg, MD, USA), according to the manufacturer’s guidance. After incubating at 37 °C in 5% CO_2_ for 48 h, the supernatants containing viruses were collected and used to infect to RAW-D cells. RAW-D cells expressing GFP or GFP-Rab11A were selected by puromycin (5 μg/mL) in MEMα containing 10% FBS, and the medium was refreshed every 3 days. After 2 weeks of culture, the puromycin-resistant cloned cells were obtained.

### 2.9. Flow Cytometry Analysis

The osteoclasts derived from RAW-D cells were used after 2 days of RANKL stimulation. The cell suspension (1 × 10^6^ cells/100 μL) incubated with the primary antibodies diluted with PBS solution containing 1% normal goat serum was put on ice for 40 min. After successive pre-incubation of the samples with Fc Block (anti-mouse CD16/CD32 antibody, #101301, Biolegend, CA, USA), specific antibodies against c-fms (anti-mouse CD115-PE-conjugated, #135505, Biolegend, CA, USA) or RANK (anti-mouse CD265-PE-conjugated, #119805, Biolegend, CA, USA) or TfR (anti-mouse CD71-PE-conjugated, #113807, Biolegend, CA, USA) or isotype control (rat IgG2a κ-PE-conjugated, #400507, Biolegend, CA, USA) were reacted on ice for 10 min, flow cytometric analyses were carried out using a MACSQuant 2.5 (Miltenyi Biotec, Tokyo, Japan).

### 2.10. Surface Biotinylation Assay

RAW-D cells (5 × 10^5^ cells) were seeded and grown in 10 cm dishes for 3 days upon RANKL stimulation. Cells were washed twice by PBS and subsequently incubated for 1 h at 4 °C with 3.0 mg/mL Sulfo-NHS-SS-Biotin (Pierce) dissolved in DPBS+. Cell dishes were rinsed in 100 mM glycine (10 min, 3×), and subsequently in 20mM glycine (10 min, 3×), both in DPBS+. Cells were harvested and lysed by the buffer LB3 containing 50 mM Tris/HCl (pH 7.4), 150 mM NaCl, 1 mM EDTA, 1% (*w*/*v*) Triton X-100, and protease inhibitor. The cell lysates were gently rotated for 1 h at 4 °C. The cell lysates were rotated by a rotator overnight at 4 °C with 40 μL Ultra Link Immobilized NeutrAvidin protein (Pierce). Followed by the incubation, beads were washed 1× with lysis buffer LB3, 2× with LB2 (LB3 not containing protease inhibitor), 2× with SWS containing 0.1% Triton X-100 in PBS (pH 7.4), 350 mM NaCl and 1 mM EDTA, and 1× with LB1 (LB2 not containing 1% (*w*/*v*) Triton X-100). Then, the beads were completely mixed with 6× sample loading buffer and boiled for 5 min before being loaded on SDS-PAGE gels.

### 2.11. CellTiter-Glo Viability Assay (CTG)

Cytotoxicity evaluation was carried out using the CellTiter-Glo Lyminescent Cell Viability Assay Kit (Promega, Madison, WI, USA), according to the manufacturer’s instructions. In total, 5 × 10^3^ cells/well were seeded and grown in 96-well white flat-bottomed plates. The plates were incubated at 37 °C in 5% CO_2_ for 24h before RANKL (300 ng/mL) addition. The plates were incubated for 3 days in 5% CO_2_, then simultaneously added with cyclohexamide (CHX) (20 μg/mL) and with or without CLQ (10 μM) to each well, and subsequently incubated for another 3 h before quenched with CellTiter-Glo^®^ (Promega, Madison, WI, USA, 50 μL/well), then centrifuged at 1000 rpm for 1 min and incubated at RT for 15 min. Luminescence was recorded with a plate reader (Molecular Devices, San Jose, CA, USA).

### 2.12. Statistical Analysis

Statistical significance was calculated using JMP Pro 15 and Microsoft Excel. Three or more mean values were compared using one-way analysis of variance (ANOVA), while comparisons of two were done with an unpaired Student’s *t*-test. *p* < 0.05 was considered to indicate statistical significance. Data were expressed as mean ± SD.

## 3. Results

### 3.1. Rab11A is Upregulated at a Late Stage of Osteoclast Differentiation

To investigate whether Rab11A was involved in osteoclastogenesis, we firstly assessed the homeostatic modification of Rab11A and two key transcription factors, c-fos and NFATc-1, essential for osteoclast differentiation [4], over a time course of RANKL stimulation. Our data showed that the endogenous levels of c-fos and NFATc-1 were transiently increased on day 1 and drastically decreased on day 3 whereas that of Rab11A was significantly increased on days 3 and 4 in RAW-D cells (Figure 1A,C) and BMMs (Figure 1B,D). More importantly, by TRAP staining, we observed that the mature osteoclasts were formed from day 3 in RAW-D cells (Figure 1E) and BMM cells (Figure 1F). Together, these results indicate that Rab11A is strongly increased at a late stage of osteoclast differentiation.

### 3.2. Rab11A Silencing Promotes Osteoclast Differentiation

To investigate if Rab11A was functionally involved in osteoclast differentiation, we first examined the effect of siRNA-mediated Rab11A silencing on osteoclast differentiation. The siRNA-induced knockdown efficacy of Rab11A was assessed by RT-qPCR and WB on days 0 and 3 of RANKL stimulation. On day 0, our results showed the remarkable reductions of Rab11A mRNA and protein levels by 99.5% and by ≈80%, respectively (Figure 2A,C), and on day 3 of RANKL treatment, by 82% and ≈ 90%, respectively (Figure 2B,D), as compared to the Ctrl si-treated groups. In TRAP staining, we observed that Rab11A silencing markedly promoted the formation of multinucleated cells (MNCs) in size and number in RAW-D-derived osteoclasts (Figure 2E–G) as well as BMM-derived osteoclasts (Figure 3A–C). Noticeably, after assessing and comparing the bone resorption area, the Rab11A-silenced osteoclasts derived from BMMs exhibited a considerable elevation in the bone-resorbing activity in comparison with that of control group (Figure 3D,E). Altogether, these findings strongly indicate a stimulatory effect of Rab11A silencing on osteoclast differentiation.

### 3.3. Rab11A Overexpression Attenuates Osteoclast Differentiation

In order to fully understand the regulatory function of Rab11A on osteoclast differentiation, we next sought to assess the effects of Rab11A overexpression on osteoclast differentiation by using RAW-D cells stably expressing GFP or GFP-Rab11A, referred to as control or Rab11A overexpression, respectively. Initially, we confirmed the expression of GFP and GFP-Rab11A (Figure 4A) in RAW-D cells cloned by puromycin selection by WB analysis. We next evaluated the formation of osteoclasts expressing GFP or GFP-Rab11A by TRAP staining. Our data showed that Rab11A overexpression strongly attenuated the formation of MNCs in size as well as in number (Figure 4B–D). Importantly, Rab11A-overexpressing osteoclasts exhibited a marked reduction in the bone-resorbing activity as compared to that of control group (Figure 4E,F). Taken together, our results suggest an inhibitory function of Rab11A overexpression on osteoclast differentiation.

### 3.4. Rab11A is Localized in Early and Late Endosomes, but not Lysosomes, and Rab11A Overexpression Triggered a Size-Based Enlargement of Early Endosomes

Previously it has been shown that Rab11A is predominantly enriched in ERCs and crucial for the regulation of the recycling of endocytosed cargos including cell surface receptors such as TfR [12,28]. Therefore, we examined the colocalization of GFP-Rab11A with several organelles-specific markers including Rab5 (early endosomes), Rab7 (late endosomes), and LAMP1 (lysosomes). Our results showed that Rab11A was localized in early and late endosomes, but not lysosomes in RAW-D cells (Figure 5A, arrowheads) and RAW-D-derived osteoclasts (Figure 5B, arrowheads). Furthermore, we observed that Rab11A overexpression caused the enhanced fluorescent intensity and a size-based enlargement of early endosomes in RAW-D cells (Appendix A), suggesting that Rab11A was engaged in regulation of accumulation of internalized cargos in early endosomes. Collectively, our data clearly indicate Rab11A vesicular localization in early and late endosomes, but not lysosomes.

### 3.5. Rab11A Silencing Upregulated the Surface Levels of c-fms and RANK Receptors

To elucidate the mechanism by which Rab11A negatively regulates osteoclast differentiation, we examined the endogenous levels of c-fms, RANK, and NFATc1 in Rab11A-silenced RAW-D cells stimulated with RANKL for 0 or 3 days. Our results showed Rab11A silencing markedly increased the endogenous levels of c-fms, RANK, and NFATc-1 proteins in RAW-D cells stimulated with RANKL for 0 day (Figure 6A–C, left panels) and for 3 days (Figure 6A–C, right panels). The same effects were also observable in BMMs (Figure 6B–D). Next, we examined if Rab11A silencing affected the surface levels of c-fms and RANK receptors by surface biotinylation assay. Interestingly, we found that Rab11A silencing markedly elevated their surface levels in RAW-D cells, following RANKL stimulation for 3 days (Figure 6E). In combination with our above findings (Figure 2 and Figure 3), our results clearly substantiate an inhibitory effect of Rab11A on osteoclast differentiation mainly via enhancing the surface abundance of c-fms and RANK receptors.

A previous study revealed a crucial role of Rab11A in regulating the trafficking of TfR to surface membrane in the polarized cells [12]. We therefore tested TfR as an indicator to investigate the transport route of Rab11A silencing-mediated c-fms and RANK receptors by flow cytometric assay. Surprisingly, we observed that Rab11A silencing markedly decreased surface level of TfR in RAW-D cells stimulated with RANKL for 2 days (Figure 6F). These results indicate that the Rab11A is obviously engaged in regulating the transport route of c-fms and RANK receptors, probably distinct from that of TfR.

### 3.6. Rab11A Overexpression Downregulated Surface Levels of c-fms and RANK Receptors in Osteoclasts

Next, we examined the mechanistic effects of Rab11A overexpression on regulation of surface levels of c-fms and RANK receptors in RAW-D cell-derived osteoclasts. Consistently, our results showed that Rab11A overexpression strongly decreased the endogenous levels of c-fms, RANK, and NFATc-1 in RAW-D cells stimulated with RANKL for 0 or 3 days (Figure 7A,B); more importantly, abolishing the surface levels of c-fms and RANK receptors in RAW-D cells following RANKL stimulation for 3 days (Figure 7C). Besides, the surface level of TfR was slightly altered by Rab11A overexpression in RAW-D cells stimulated with RANKL for 2 days (Figure 7D). These data further clarify (i) a suppressive role of Rab11A overexpression on osteoclast differentiation through weakening the surface abundance of c-fms and RANK receptors and suggest that (ii) Rab11A overexpression-mediated transport route of c-fms and RANK receptors distinct from that of TfR.

### 3.7. Rab11A Overexpression Facilitated Lysosome-Induced Degradation of c-fms and RANK Receptors in RAW-D Cell-Derived Osteoclasts

So far, it has been well-characterized that lysosome-degraded surface receptors are processed from early endosomes to late endosomes before transported to lysosomes for proteolysis [29,30,31]. As above (Figure 5), we observed the vesicular localization of Rab11A in early and late endosomes, we therefore hypothesized that Rab11A might have functionally engaged in the early and late endosome-mediated delivery of c-fms and RANK surface receptors to lysosomes in osteoclasts. To address this hypothesis, we first examined the effects of Rab11A overexpression on lysosomal activity by assessing the expression levels of the lysosomal marker, LAMP1 and two major lysosomal enzymes, Cathepsins B and D. By Rab11A overexpression, the endogenous levels of LAMP1, Cathepsins B and D were insignificantly altered with RANKL stimulation for 0 days while strongly increased with RANKL stimulation for 3 days (Figure 8A,B), suggesting a functional connection of Rab11A overexpression to augmentation of lysosomal activity in osteoclasts. Next, we used CLQ, a specific blocker of lysosomes, and CHX, a specific blocker of newly protein synthesis, to investigate if lysosomes were engaged in Rab11A-mediated regulation of c-fms and RANK protein. Surprisingly, it was observable that lysosomal inhibition enhanced the endogenous levels of c-fms and RANK protein (Figure 8C,D), regardless of Rab11A overexpression, suggesting that lysosomes functionally contributed to proteolytic degradation of c-fms and RANK proteins in osteoclasts. More interestingly, Rab11A overexpression also triggered lysosome-induced degradation of c-fms and RANK (Figure 8C,D). There was no significant toxic effect observed in all the cases treated (Figure 8E). These findings elucidate an important function of lysosomes on proteolytically degrading c-fms and RANK surface receptors, and this process would be facilitated by Rab11A overexpression in osteoclasts, thereby weakening osteoclast differentiation.

## 4. Discussion

Rab GTPase proteins are localized in various intracellular compartments, including endosomes, Golgi complex, lysosomes, and the cell surface. The Rab GTPases are master regulators of distinct steps of intracellular vesicle transport, protein trafficking, membrane targeting, and fusion in eukaryotic cells [32]. In principle, Rab GTPase proteins are thought to be functionally interconnected to one another to regulate cargo transport routes through intracellular compartments [33], such as from recycling endosomes to cell surface and/or to cytosolic organelles in various cell types, functions especially active in cancer cells [34]. In our current study, we initially found Rab11A was upregulated as mature osteoclasts were formed (Figure 1), conjecturing that Rab11A possibly served as a negative regulator of osteoclastogenesis. Indeed, Rab11A silencing strongly promoted osteoclast formation, bone-resorbing activity, augmented surface levels of c-fms and RANK receptors, resulting in NFATc-1 upregulation in both osteoclasts derived from RAW-D cells (Figure 2E–G) and BMMs (Figure 3C–E). Besides, Rab11A overexpression reduced such characteristics in RAW-D cell-derived osteoclasts (Figure 4B–F). From these findings, it was clarified an inhibitory role of Rab11A for regulation of osteoclastogenesis. Though our bone resorption data indicated the bone-resorbing activity of osteoclasts derived from both RAW-D cells and/or BMMs, it would be noteworthy of examining the other factors such as actin ring formation, Cathepsin K (CTSK), DC (or OC)-STAMP, and/or ATPase, H^+^ transporting, lysosomal pump (atp6v0d2) expression, in order to fully understand the physiological role of Rab11A in osteoclast differentiation regulation. Therefore, it would be probably one of our future research goals.

The vesicular cellular localization of Rab11A was observed in early and late endosomes in both premature (Figure 5A) and mature (Figure 5B) osteoclasts, and Rab11A overexpression caused the enhanced fluorescent intensity and the size-based enlargement of early endosomes in RAW-D cells (Appendix A). Interestingly, our previous report also showed the same effects of Rab44 on enlarging early endosomes in RAW-D cells [21]. From these observations, we therefore proposed one possibility that Rab11A overexpression promoted the accumulation of the internalized cargos including the surface receptors, in early endosomes though this phenomenal consequence has been obscure, and needs to be further investigated. Of note, an earlier report has revealed endogenous depletion of Rab11A caused the enlargement of early endosomes, and augmentation of late endosomal and lysosomal activities in nonpolarized HeLa cells [35], suggesting a functional complexity of Rab11A in human cells.

So far, there have been two endosomal recycling pathways, comprising fast and slow recycling routes, in which TfR receptor was elucidated to recycle to the cell surface exclusively via the slow recycling route [12]. As described herein, Rab11A silencing strengthened the surface levels of c-fms and RANK receptors (Figure 6E) while inhibiting that of TfR (Figure 6F). Reversely, Rab11A overexpression decreased the cell surface levels of c-fms and RANK receptors (Figure 7C) while slightly increasing that of TfR receptor (Figure 7D). Based on these observations, we proposed that Rab11A-mediated transport route of c-fms and RANK surface receptors is distinct from that of TfR. By the method of exclusion, we hypothesized that c-fms and RANK surface receptors might have been recycled to surface membrane through the fast recycling route. On the contrary, the surface levels of c-fms and RANK receptors were markedly decreased with respect to Rab11A overexpression in RAW-D cell-derived osteoclasts (Figure 7C), suggesting that the Rab11A-mediated transport route of c-fms and RANK surface receptors was not through the fast recycling route. On the emphasis of the functional importance of lysosomes in regulating proteolysis of cellular cargos such as surface receptors via the axis of early endosome–late endosomes–lysosomes, it prompted us to examine if Rab11A overexpression might have been associated with lysosomal activity. Indeed, Rab11A overexpression markedly surged the lysosomal activity through upregulation of LAMP1, and Cathepsins B and D in osteoclasts (Figure 8A). Intriguingly, using a specific blocker of lysosomes, CLQ, we found that lysosomal inhibition elevated the endogenous levels of c-fms and RANK proteins (Figure 8C,D), regardless of Rab11A overexpression, upon RANKL stimulation, suggesting a natural characteristic of lysosomes degrading c-fms and RANK proteins in osteoclasts. Additionally, the same effects were found in Rab11A-overexpressing RAW-D cell-derived osteoclasts (Figure 8C,D).

Our study touches upon a Rab11A-specific mechanism on potential bone resorption protection. An early report did clarify the function of Rab11 on regulating the surface proteins, comprising the surface receptors [36]. Intriguingly, we unmasked a regulatory role of Rab11A silencing in osteoclastogenesis via limiting the cell surface levels of c-fms and RANK receptors both of which are prerequisite for osteoclast differentiation (Figure 6). Besides, our earlier report obviously disclosed the functional roles of Rab27A as the negative modulator of osteoclast differentiation, using Rab27A-deficient Ashen mice [20]. The osteoclasts originating from these mice were much bigger in size than those from of the wild-type counterparts, of note, similar effects were also observable by Rab11A silencing in the RAW-D cell-derived osteoclasts. In addition to the abovementioned effect of Rab27A on osteoclast differentiation, it was clear Rab27A was identified to regulate the transport of lysosomal system-related organelles to the ruffled border in collaboration with Rab7 [37], thereby suggesting an inhibitory effect of the bone resorption activity in Rab27-deficient osteoclasts. Furthermore, one of the Rab family proteins, Rab44, was functionally elucidated to be a suppressive modulator of osteoclast differentiation via effecting on the intracellular Ca^2+^ level, thereby modulating NFATc1 signaling cascades [21]. Structurally, Rab44 is an atypical Rab-GTPase containing several additional domains such as the EF-hand domain, coiled-coil domain, and the Rab-GTPase domain as well [38]. Of which, EF-hand domain highly conserved in vertebrates such as human, mouse, and rat, was the regulator of intracellular Ca^2+^ level. Nonetheless, although Rab11A, a typical Rab-GTPase lacking in the EF-hand domain, was thought to not be involved in Ca^2+^ oscillation, it should be further investigated.

## 5. Conclusions

Our study revealed a novel role of Rab11A serving as a negative regulator of osteoclast differentiation mainly via weakening the surface abundance of c-fms and RANK receptors, whose activation results in upregulating NFATc1 signaling cascades, thereby inducing osteoclast differentiation. More intriguingly, our study was also the first report reflecting one another function of lysosomes on proteolytically degrading c-fms and RANK receptors in osteoclasts, causally resulting in abolishing osteoclast differentiation. Altogether, our novel findings extend our knowledge of the regulatory mechanisms of osteoclast characteristics and bone physiology, probably facilitating development of establishment of the new therapeutic drugs effective to the treatments of both inflammation-caused bone loss and macrophage-triggered inflammatory syndromes.

## Figures and Tables

**Figure 1 cells-09-02384-f001:**
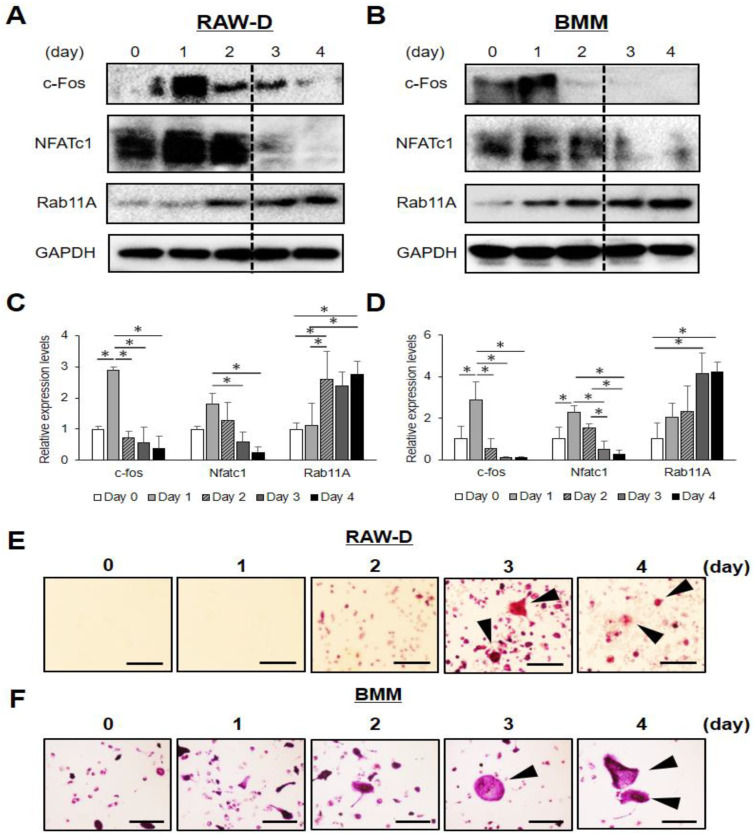
Rab11A upregulation at a late stage of osteoclast differentiation. (**A**,**B**) RAW-D cells (**A**) or BMMs (**B**) were treated with RANKL over the indicated time course. Total expression levels of c-Fos, NFATc-1, Rab11A, and GAPDH used as a loading control were evaluated by WB. (**C**,**D**) Quantitative analyses of Western blot for c-fms and RANK, NFATc1, in RAW-D cells (**C**) or BMMs (**D**). GAPDH was used as an internal control. * *p* < 0.05, (**E**,**F**) TRAP staining was carried out to assess the formation of mature osteoclasts differentiated from RAW-D cells (**E**) or from BMMs (**F**) upon RANKL stimulation over a time course. Arrowheads indicated the mature osteoclasts. Scale bars: 200 μm. Data shown were the representative of three independent experiments.

**Figure 2 cells-09-02384-f002:**
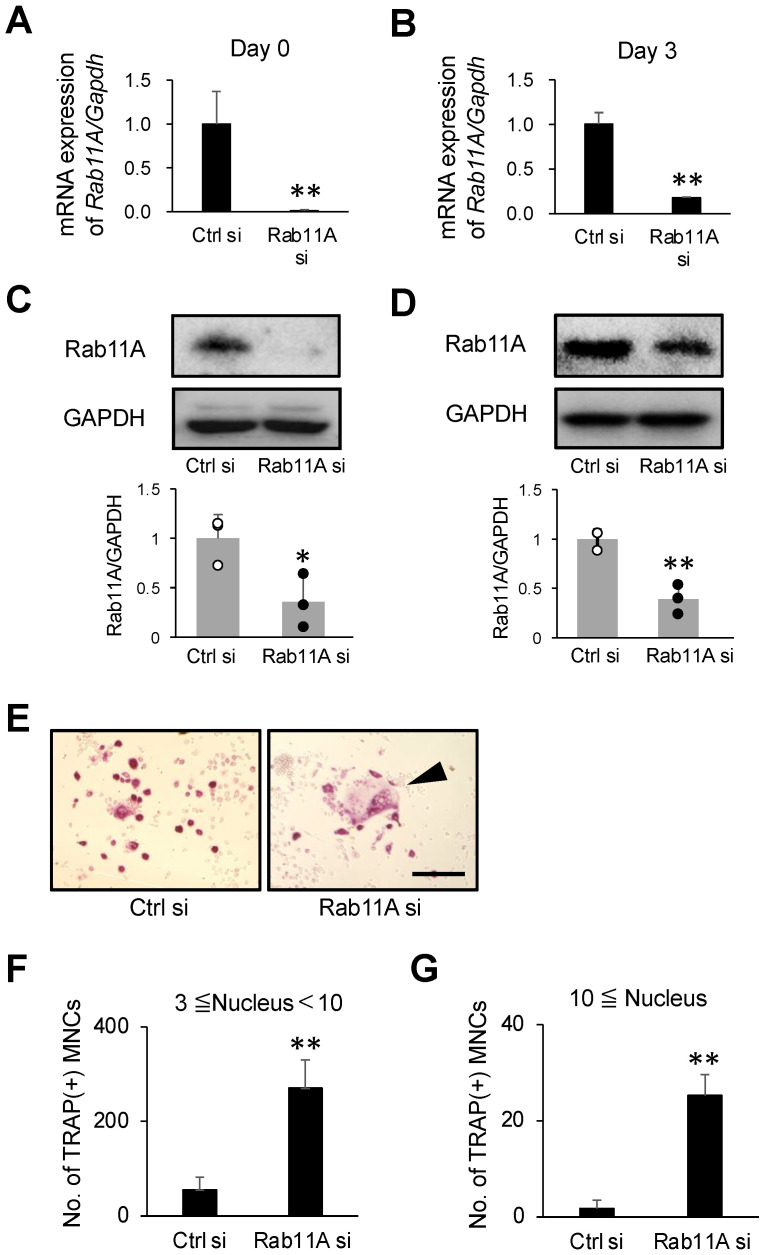
The effects of Rab11A silencing on osteoclast differentiation. (**A**) RAW-D cells were transfected with nontargeting (Ctrl) or Rab11A-specific siRNA for 24 h without RANKL stimulation. The knockdown efficacy of Rab11A mRNA was analyzed by qRT-PCR. (**B**) RAW-D cells were transfected with Ctrl si or Rab11A si for 24 h, followed by RANKL stimulation for 3 days. The knockdown efficacy of Rab11A mRNA levels was analyzed by RT-qPCR. (**C**,**D**) Upper: The endogenous level of Rab11A protein was assessed by WB on day 0 (**C**) and day 3 (**D**). Lower: Column scatter plotting to compare Rab11A protein level on day 0 (**C**) and day 3 (**D**). (**E**) TRAP staining of Ctrl si or Rab11A si-treated osteoclasts. The cells were treated with Ctrl si or Rab11A si 24 h, followed by RANKL stimulation for 3 days. An arrowhead indicated the mature osteoclasts. Scale bars: 200 μm. (**F**,**G**) The number of TRAP-positive osteoclasts with 3–10 nuclei (**F**), or with more than 10 nuclei (**G**) per viewing field was counted. * *p* < 0.05, ** *p* < 0.01, *n* = 3. Data are the representative of three independent experiments.

**Figure 3 cells-09-02384-f003:**
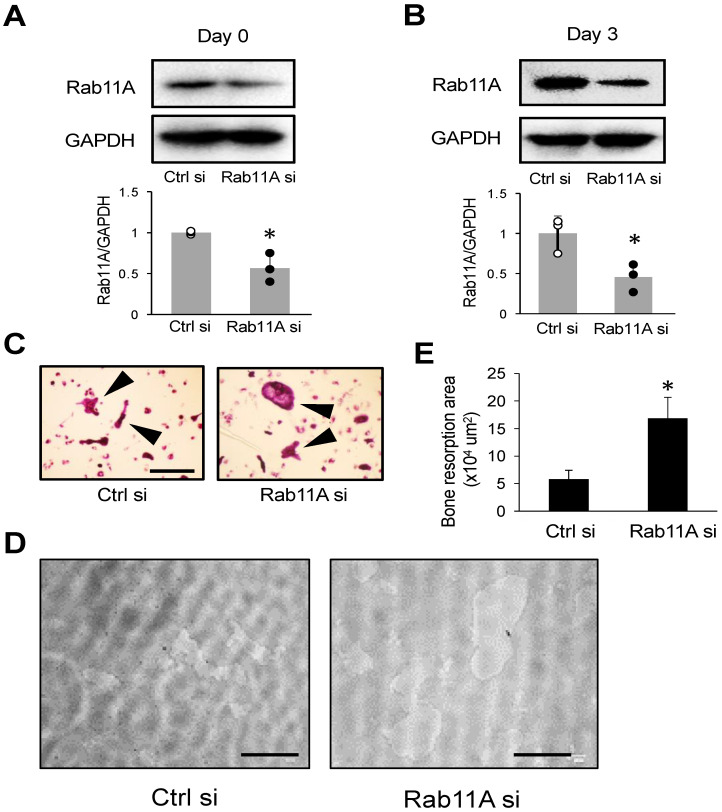
The effects of Rab11A silencing on BMM-derived osteoclast differentiation. (**A**) Upper: BMM cells were transfected with nontargeting siRNA (Ctrl si) or Rab11A-specific siRNA (Rab11A si) for 24 h without RANKL stimulation. Lower: Column scatter plotting to compare Rab11A protein level. (**B**) Upper: BMM cells were transfected with Ctrl siRNA or Rab11A siRNA for 24 h, followed by RANKL stimulation for 3 days. The endogenous level of Rab11A protein was evaluated by Western blotting. Lower: Column scatter plotting to compare Rab11A protein level. (**C**) TRAP staining of osteoclast transfected Ctrl si or Rab11A si. Arrowheads showed large osteoclasts. Bars: 200 μm. (**D**) Images of the bone resorption area of BMM-derived osteoclasts transfected with Ctrl si or Rab11A si. Bars: 100 μm. (**E**) The resorption area was determined using Image J software. The data are represented as mean ± SD of values from three independent experiments. * *p* < 0.05, compared to control cells.

**Figure 4 cells-09-02384-f004:**
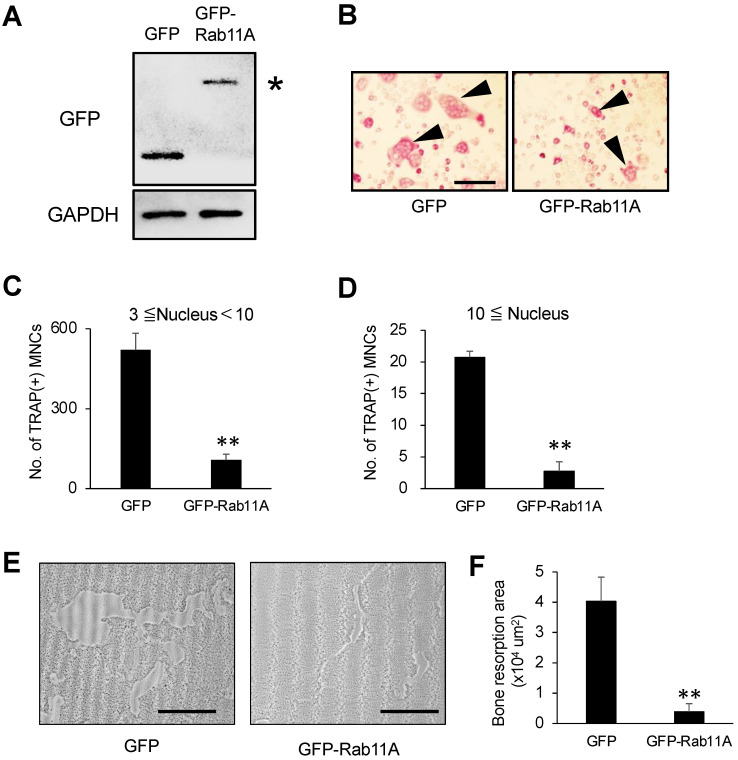
The effect of Rab11A overexpression on osteoclastogenesis. (**A**) GFP or GFP-Rab11A expression was determined in RAW-D cells transduced with either retrovirus vector encoding GFP or GFP-tagged Rab11A (GFP-Rab11A). An “asterisk” indicated the GFP-Rab11A band. (**B**) TRAP staining of GFP and GFP-Rab11A-expressing osteoclasts derived from RAW-D cells. Arrowheads showed mature osteoclasts. Scale bars: 200 μm. (**C**,**D**) The number of TRAP-positive multinucleated osteoclasts with 3–10 nuclei or more than 10 nuclei per viewing field was counted. ** *p* < 0.01. (**E**) Images of the bone resorption area of RAW-D-derived osteoclasts expressing GFP-alone or GFP- Rab11A. Scale bars: 100 μm. (**F**) The resorption area was determined using Image J software. The data are represented as mean ± SD of values from three independent experiments.

**Figure 5 cells-09-02384-f005:**
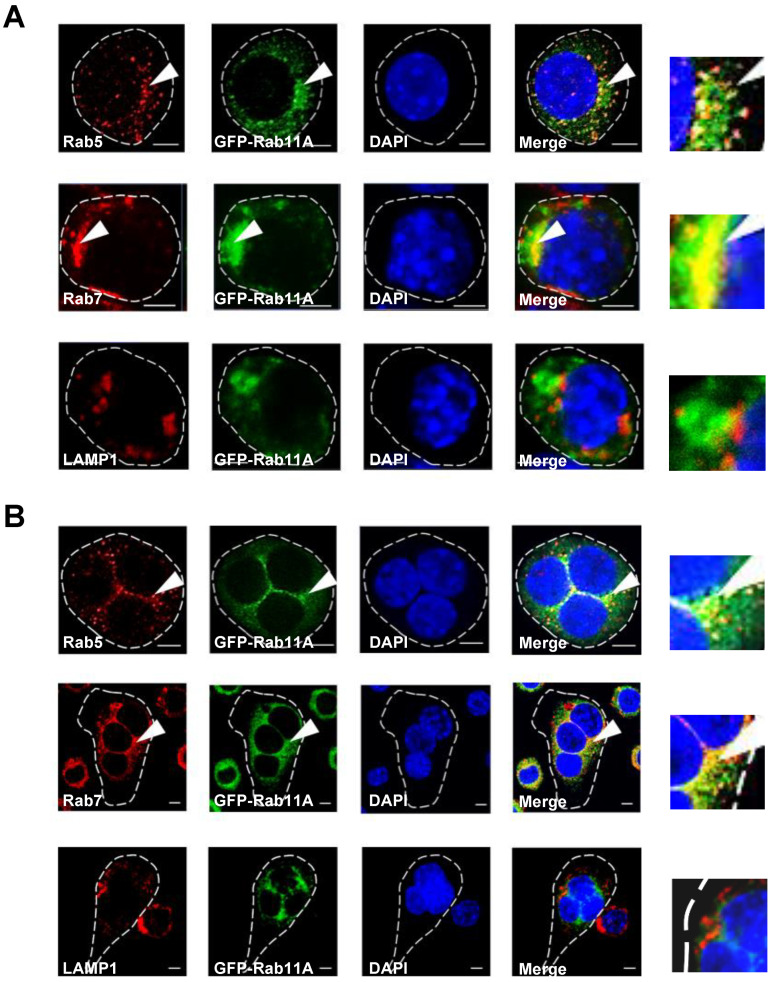
Subcellular localization of Rab11A in RAW-D cells and RAW-D cell-derived osteoclasts expressing GFP-Rab11A (green). (**A**,**B**) RAW-D cells (**A**) or osteoclasts following RANKL stimulation for 3 days (**B**) were seeded on cover glasses, permeabilized with 0.2% Triton X-100 diluted in PBS, subsequently reacted with one of the antibodies against Rab5, Rab7, or LAMP1 (red, as indicated). DNA was stained with DAPI (blue). Arrowheads indicated the positive region of GFP-Rab11A and each organelles-specific markers. Scale bars: 5 μm. The images shown were the representative of three independent experiments.

**Figure 6 cells-09-02384-f006:**
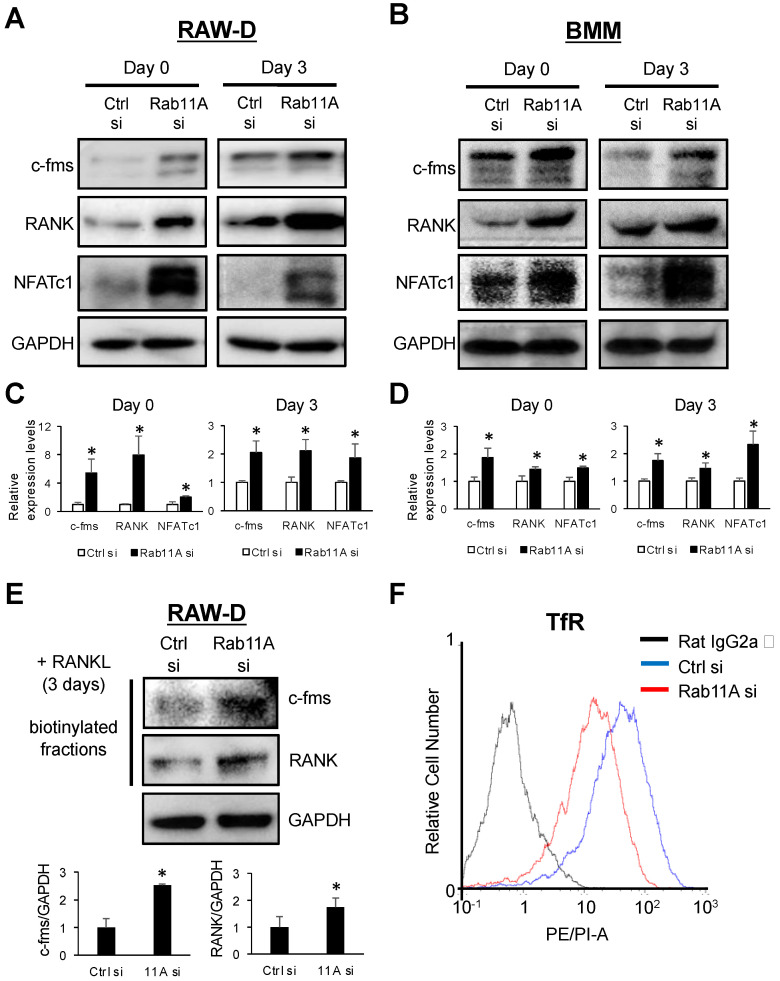
The effects of Rab11A silencing on cell surface levels of c-fms and RANK receptors in RAW-D cells stimulated with RANKL (300 ng/mL). (**A**) The endogenous levels of c-fms, RANK, and NFATc1 in Ctrl or Rab11A si-treated RAW-D cells stimulated with RANKL for 0 (left panels) or 3 days (right panels). (**B**) The endogenous levels of c-fms, RANK and NFATc1 in Ctrl or Rab11A si-treated BMMs stimulated with RANKL for 0 (left panels) or 3 days (right panels). (**C**,**D**) Quantitative analyses of Western blot for c-fms, RANK, NFATc1, in RAW-D cells (**C**) or BMMs (**D**). GAPDH was used as an internal control. * *p* < 0.05. (**E**) Upper: The biotinylated fractions were subjected to immunoblotting with anti-mouse c-fms and anti-mouse RANK antibodies, and whole cell lysates (WCLs) were subjected to immunoblotting with anti-GAPDH antibody as a loading control in RAW-D cells following RANKL stimulation for 3 days. Lower: Quantitative analyses of Western blot for c-fms, RANK. * *p* < 0.05. (**F**) Flow cytometric analyses of TfR using Ctrl or Rab11A siRNA-transfected RAW-D cells, followed by RANKL stimulation for 2 days. Data are representative of three independent experiments.

**Figure 7 cells-09-02384-f007:**
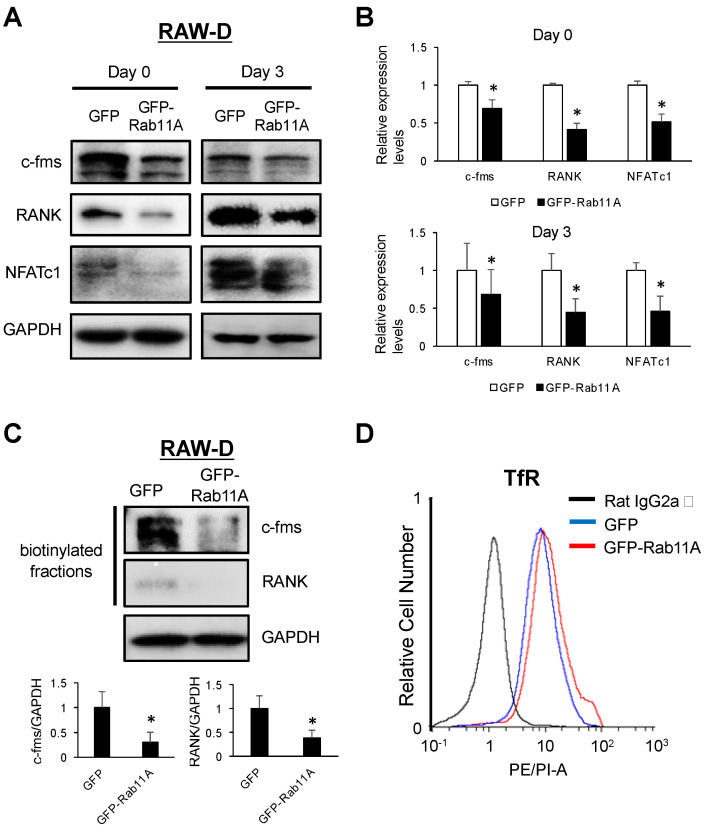
The effects of Rab11A overexpression on cell surface levels of c-fms and RANK receptors in RAW-D cells upon RANKL stimulation. (**A**) The endogenous levels of c-fms, RANK, and NFATc1 in RAW-D cells expressing GFP (control) or GFP- Rab11A following RANKL stimulation for 0 or 3 days. (**B**) Quantitative analyses of Western blot for c-fms, RANK, NFATc1. GAPDH was used as an internal control. * *p* < 0.05. (**C**) Upper: The biotinylated fractions were subjected to immunoblotting with anti-mouse c-fms and anti-mouse RANK antibodies, and WCLs were subjected to immunoblotting with GAPDH-HRP antibody as a loading control in RAW-D cells expressing GFP (control) or GFP-Rab11A. Lower: Quantitative analyses of Western blot for c-fms, RANK. (**D**) Flow cytometric analyses of TfR were done using RAW-D cells expressing GFP (control) or GFP-Rab11A. Data are the representative of three independent experiments.

**Figure 8 cells-09-02384-f008:**
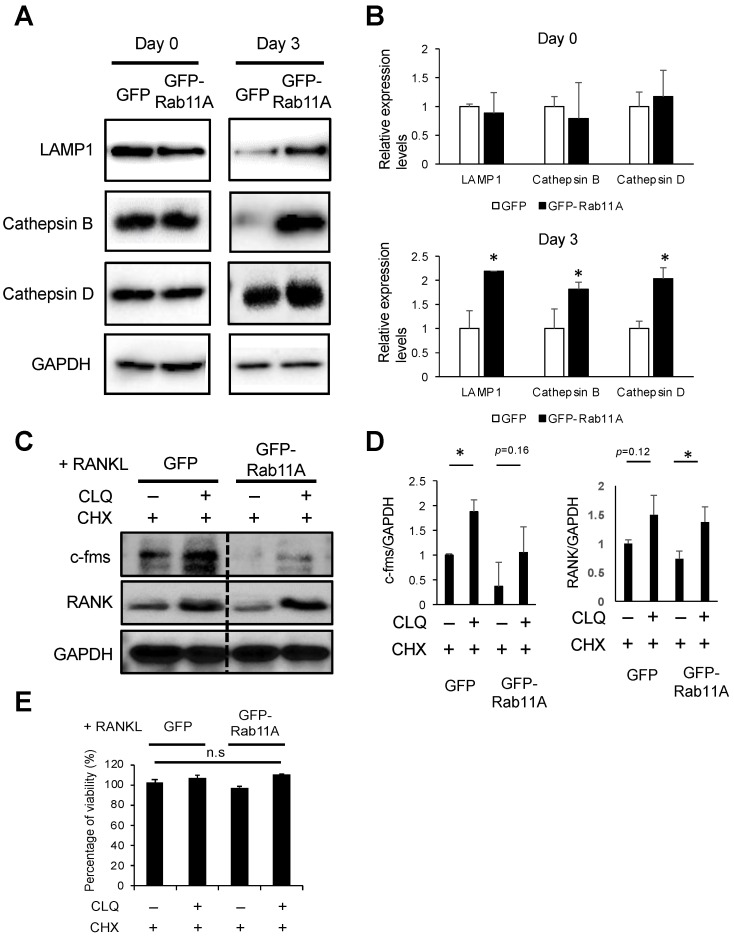
The lysosomal function on c-fms and RANK protein degradation ameliorated by Rab11A overexpression in RAW-D cell-derived osteoclasts. (**A**) The endogenous levels of LAMP1 (a specific lysosomal receptor), and Cathepsins B and D (two major lysosomal enzymes) in GFP or GFP-Rab11A-expressing RAW-D cells following RANKL stimulation for 0 (left panel) or 3 days (right panel) were analyzed by WB with anti-rat LAMP1, anti-rabbit Cathepsins B and D, and GAPDH-HRP (loading control). (**B**) Quantitative analyses of Western blot for LAMP1, Cathepsin B, Cathepsin D. GAPDH was used as an internal control. * *p* < 0.05. (**C**) After 3 days of RANKL (300 ng/mL) stimulation, RAW-D cell-derived osteoclasts were treated simultaneously with 20 μg/mL CHX and with or without 10 μM CLQ for 3 h. c-fms and RANK protein levels were assessed by WB with anti-mouse c-fms, anti-mouse RANK, and GAPDH-HRP (loading control) antibodies. (**D**) Quantitative analyses of Western blot for c-fms and RANK. GAPDH was used as an internal control. * *p* < 0.05. (**E**) Cell viability was assessed by the cellular ATP content measurement using the CTG Assay system. After stimulated with RANKL (300 ng/mL) for 3 days, RAW-D cells were subsequently added with 20 μg/mL CHX and with or without 10 μM CLQ for 3 h. The values were the average of triplicate determinations with the S.D indicated by error bars. n.s; no significant.

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
