# Peer review of "Rab11A Functions as a Negative Regulator of Osteoclastogenesis through Dictating Lysosome-Induced Proteolysis of c-fms and RANK Surface Receptors"

_cells, 2020, doi:10.3390/cells9112384_

Round 1
Reviewer 1 Report
This manuscript by Okusha et al. describes the role of Rab11A on osteoclastogenesis via lysosome-induced proteolysis of c-fms and RANK receptors. Authors show that Rab11A attenuates osteoclast differentiation, resorption activity, while its partial silencing by siRNA promotes them. They further show that Rab11a colocalizes with early and late endosomes, and that Rab11A overexpression enhances lysosome-induced degradation of c-fms and RANK receptors. This study is very well designed and performed. Below are my very minor comments.
- This study uses chloroquine (CLQ) as a specific blocker of lysosomes. This is known to neutralize the lysosomal pH. Please include a description on the working mechanism and why this drug is used.
- In Fig. 1 C and D, please clearly describe which groups are compared for statistical analysis.
- In Fig. S2, include a detailed descripton on quantifying area/size since this quantification might be highly dependent on chosen threshold values for each image.
Author Response
This manuscript by Okusha et al. describes the role of Rab11A on osteoclastogenesis via lysosome-induced proteolysis of c-fms and RANK receptors. Authors show that Rab11A attenuates osteoclast differentiation, resorption activity, while its partial silencing by siRNA promotes them. They further show that Rab11a colocalizes with early and late endosomes, and that Rab11A overexpression enhances lysosome-induced degradation of c-fms and RANK receptors. This study is very well designed and performed. Below are my very minor comments.
We appreciate this reviewer for the peer review and are delighted to respond point-by-point below.
- This study uses chloroquine (CLQ) as a specific blocker of lysosomes. This is known to neutralize the lysosomal pH. Please include a description on the working mechanism and why this drug is used.
Response 1:
There have been two well-characterized lysosomal blockers, Chloroquine (CLQ), which diffuses into lysosomes, causing alkalinization of lysosomal lumen, thereby impairing lysosomal hydrolases, and Bafilomycin A1 (BafA1), which inhibits the ability of the vacuolar type H+-ATPase (v-ATPase) to transfer protons into lysosomes and/or endosomes. v-ATPases, also localized on the ruffled border plasma membrane of bone-resorbing osteoclasts, mediated extracellular acidification for bone demineralization during bone resorption; therefore, we suggested that BafA1 was not a potent lysosomal blocker for the osteoclastogenesis-related studies. Ultimately, we selected CLQ to carry out the designated experiments. As agreeing with the reviewer’s suggestion, we added the detailed information of CLQ functions on lysosomal inhibition in the “Introduction Section” (P2L84-86). In addition, we also cited the reference for this description.
- In Fig. 1 C and D, please clearly describe which groups are compared for statistical analysis.
Response 2:
As following the reviewer’s comment, we added the detailed description in Figure legend (P6L238) of the manuscript. Also, we described the “Statistical Analysis” more obviously in the “Materials and Methods” Section (P5L218-221) as follows:
“Statistical significance was calculated using JMP pro 15 and Microsoft Excel. Three or more mean values were compared using one-way analysis of variance (ANOVA), while comparisons of two were done with an unpaired Student’s t-test. p < 0.05 was considered to indicate statistical significance. Data were expressed as Mean ± SD.”
- In Fig. S2, include a detailed description on quantifying area/size since this quantification might be highly dependent on chosen threshold values for each image.
Response 3:
As agreeing to “reviewer 3’s suggestion”, we set Fig. S1 (from the original M.S) as Fig. 3 (from the revised M.S); therefore, Fig. S2 of the original manuscript would be Fig. S1 of the revised M.S. In Fig. S1B (revised M.S), we were sorry for repeating a similar description for both graphs. Therefore, we would like to delete the right graph and added a detailed description in the figure legend (P11L326-328).
We hope that our explanations have addressed your concern as well as met your demand.
Reviewer 2 Report
In this manuscript Okusha et al defined that Rab11A negatively regulates OCLgenesis through the inhibition of the surface expression of c-fms and RANKL. They additionally showed the role of lysosomes on proteolytically degrading c-fms and RANK receptors in osteoclasts. The manuscript is well written and results are conclusive with a thorough review of literature. The findings may have implications to other skeletal disorders such as rheumatoid arthritis, periodontal disease and cancer induced bone loss. I have the following minor suggestions/comments to further improve the manuscript.
1. Authors described that Rad11A expression was upregulated during OCLgenesis Although stated in the text, in fig1A RAW-D data, the expression of Rab11a was decreased on day 1. Authors should show the consistent data.
2. In fig1B Day0 no treatment, NFATc1 is marginally expressed, but in fig5b is abundant. Authors should show the consistent data.
3. It is difficult to see the bone resorption image in FigS1D and 3E. I recommend the authors to do the von Kossa staining to make a contrast.
4. Authors showed that Rab11a regulates bone resorption capacity. Did authors test the actin ring formation or CTSK, atp6v0d2 expression or function? Authors should discuss or it is better to show the data.
5. I’m interested in the role of Rab11a on DC (or OC)-STAMP expression, since authors stated the relationship between Rab11a expression and multinucleation. If possible, authors discuss it.
6. minor comments
P2L46: decrease the space between “lysosomal” and “activity”
P3L116: add space between “50” and “ng/mL”
P4L148 Sigma Aldrich’s headquarter is not in Japan but in US.
P10L318: add space between “5” and “um”
P15L418: add space between “10” and “um”
Author Response
In this manuscript Okusha et al defined that Rab11A negatively regulates OCLgenesis through the inhibition of the surface expression of c-fms and RANKL. They additionally showed the role of lysosomes on proteolytically degrading c-fms and RANK receptors in osteoclasts. The manuscript is well written and results are conclusive with a thorough review of literature. The findings may have implications to other skeletal disorders such as rheumatoid arthritis, periodontal disease and cancer induced bone loss. I have the following minor suggestions/comments to further improve the manuscript.
We appreciate the reviewer’s comments and are delighted to respond point-by-point below.
1. Authors described that Rad11A expression was upregulated during OCLgenesis Although stated in the text, in fig1A RAW-D data, the expression of Rab11a was decreased on day 1. Authors should show the consistent data.
Response 1:
We have added the new data indicating the Rab11A expression consistency (3rd panel, Fig. 1A), based on the reviewer’s suggestion. In fact, there was an insignificant alteration of Rab11A expression levels between day 0 and day 1 upon RANKL stimulation.
- In fig1B Day0 no treatment, NFATc1 is marginally expressed, but in fig5b is abundant. Authors should show the consistent data.
Response 2:
Experimentally, we could detect a relatively high expression level of NFATc-1 on day 0 for the siCtrl-treated BMMs. However, fully agreeing to the reviewer’s suggestion on making the data consistency, we have added the new data indicating the expression level of NFATc-1 (Fig. 1B, 2nd panel), which is consistent with that of Fig. 5B (Fig. 6B of the revised M.S).
- It is difficult to see the bone resorption image in FigS1D and 3E. I recommend the authors to do the von Kossa staining to make a contrast.
Response 3:
We appreciate the reviewer’s recommendation, and it would be probably our future work strategy to improve the quality of bone resorption images. To improve our current bone resorption images (Fig. S1D and Fig. 3E of the original M.S, which are corresponding to Fig. 3D and Fig. 4E of the revised M.S, respectively), we have carried out the following steps:
(i) we deleted the dot lines
(ii) we modified the image background clearer and brighter than the last ones for easier observation.
- Authors showed that Rab11a regulates bone resorption capacity. Did authors test the actin ring formation or CTSK, atp6v0d2 expression or function? Authors should discuss or it is better to show the data.
Response 4:
We thank the reviewer for a good suggestion. Though we did not examine Rab11A-caused actin ring formation, CTSK, or atp6v0d2, Rab11A silencing or overexpression did alter osteoclast formation and bone-resorbing activity. Therefore, we speculate that Rab11A endogenous alternation potentiates the consequent modification of actin ring formation, CTSK, and/or atp6v0d2. Certainly, it would be our future work plan to further elucidate the physiological role of Rab11A for osteoclast differentiation regulation. In this study, we would like to refer it to the “discussion” section (P16L438-443) as our future work plan.
- I’m interested in the role of Rab11a on DC (or OC)-STAMP expression, since authors stated the relationship between Rab11a expression and multinucleation. If possible, authors discuss it.
Response 5:
It is supposed to be a good point for further strengthening our findings. As our observations indicating the inhibitory role of Rab11A in regulating osteoclast formation, bone-resorbing activity, we speculate that Rab11A overexpression would alleviate the expression level of DC (or OC)-STAMP while its endogenous silencing enhanced that of DC (or OC)-STAMP. However, whether Rab11A endogenous alternation is directly associated with the endogenous turnover and/or transport of DC (or OC)-STAMP has been completely obscure. Therefore, it would be one of our future work plans to completely elucidate the physiological role of Rab11A for osteoclast differentiation regulation. Totally agreeing to the reviewer’s suggestion, we have cited our discussion about DC (or OC)-STAMP in the “Discussion Section” (P16L440).
- minor comments 
P2L46: decrease the space between “lysosomal” and “activity”
P3L116: add space between “50” and “ng/mL”
P4L148 Sigma Aldrich’s headquarter is not in Japan but in US.
P10L318: add space between “5” and “um”
P15L418: add space between “10” and “um”
Response 6:
Thank you for your corrections. We revised these misprints.
Reviewer 3 Report
Okusha and colleagues described a new function for Rab11A as a negative regulator of osteoclastogenesis. The experiments are quite elegant and the conclusion straighforward. There are few modifications and few questions the authors should answer before the paper could be accepted for publication, but as it is, the manuscript is interesting and the data convincing.
First, the English should be corrected, there are some sentences hard to understand.
In Figure 1, it is strange that although the OC are mature there are no expression of c-Fos or NFATc1. Could the authors comment on that ? The panels C and D can’t be analyzed by a t test. This should be corrected.
The Figure S1 should be added in the Figure 1. This very important to show that what is observed in RAW cells is observed in BMM.
In Figure 4, the authors should add a high magnification of the merge image to show the non co-localization of Rab11A and Lamp1.
In figure S2, the DAPI staining should be added, for understanding purposes.
Although the panel B left is quite understandable, the right panel is very puzzling. In addition those results are not clearly presented. Moreover, in the description, it is stated that there is a enhanced intensity observed, although nothing is shown in this regard. Could this be corrected ?
Are the authors sure of the sentence beginning line 334 ? Are they really talking of a stimulatory effect of Rab11Q. Isn’t it a inhibitory effect on osteoclast differentiation.
In the Figure 5, the values of the first graph on panel C are strange, 12 really ?
In the Figure 7, Cathepsins B and D expressions are increased at D3. How the authors could explain that situation ? Is there an increased lysosome number ?
Line 414, densitometry is mentionned althought such information are not given in the figure 7.
Why, line 492 and following, the concluding remarks concern the macrophages ? It only was question of osteoclast throughout this manuscript.
Could the authors extrapolate on the way Rab11A would be expression in macrophages as they differenciate from a commun precursor in a kind of competition with osteoclasts ?
Author Response
Okusha and colleagues described a new function for Rab11A as a negative regulator of osteoclastogenesis. The experiments are quite elegant and the conclusion straightforward. There are few modifications and few questions the authors should answer before the paper could be accepted for publication, but as it is, the manuscript is interesting and the data convincing.
We appreciate this reviewer for the peer review and are delighted to response point-by-point below
- First, the English should be corrected, there are some sentences hard to understand.
Response 1:
We have revised the manuscript and included some sentences to make the manuscript more clearly. We hope that they have met your demand.
- In Figure 1, it is strange that although the OC are mature there are no expression of c-Fos or NFATc1. Could the authors comment on that?
Response 2:
As well-characterized, two master transcription factors of osteoclastogenesis, NFATc-1, and c-fos expressions are transiently and considerably increased in a range of day 0-2 of RANKL stimulation, and drastically decreased from the late of day 2, or early of day 3. From day 0 to day 3, RAW-D cells and/or BMMs are differentiated into mononuclear osteoclasts that will be undergone nuclear fusion to become mature osteoclasts. Once mature osteoclasts are formed, CTSK, TRAP, and MMP9 would be abundantly present, and secreted into the bone milieu for bone resorption; therefore, NFATc-1 and c-fos functions are less important at this stage as compared to those of the early stage of osteoclast differentiation. After all, we suggest that these phenomenal properties of c-fos and NFATc-1 expressions are the homeostatic modification thereof during osteoclast differentiation.
- The panels C and D can’t be analyzed by a t test. This should be corrected.
Response 3:
We analyzed by using ANOVA and corrected accordingly.
- The Figure S1 should be added in the Figure 1. This very important to show that what is observed in RAW cells is observed in BMM.
Response 4:
Because of the large number of data, we have shown in the M.S, we decided to set Fig. S1 of the original M.S as Fig. 3 of the revised M.S
- In Figure 4, the authors should add a high magnification of the merge image to show the non co-localization of Rab11A and Lamp1.
Response 5:
We modified and added the highly magnified image of LAMP1.
- In figure S2, the DAPI staining should be added, for understanding purposes.
Response 6:
We added the image indicating the nuclei were stained with DAPI.
- Although the panel B left is quite understandable, the right panel is very puzzling. In addition those results are not clearly presented. Moreover, in the description, it is stated that there is a enhanced intensity observed, although nothing is shown in this regard. Could this be corrected ?
Response 7:
As described in Response 4, we set Fig. S1 (from the original M.S) as Fig. 3 (from the revised M.S); therefore, Fig. S2 of the original manuscript would be Fig. S1 of the revised M.S. In Fig. S1B (revised M.S), we were sorry for repeating a similar description for both graphs. Therefore, we would like to delete the right graph and added a detailed description in the figure legend (P11L326-328).
- Are the authors sure of the sentence beginning line 334 ? Are they really talking of a stimulatory effect of Rab11Q. Isn’t it a inhibitory effect on osteoclast differentiation.
Response 8:
We are sorry for our mistyping. The word “stimulatory” should be replaced by the word “inhibitory” (P11L340).
- In the Figure 5, the values of the first graph on panel C are strange, 12 really ?
Response 9:
We did calculate again; nonetheless, we obtained the same results as we had.
- In the Figure 7, Cathepsins B and D expressions are increased at D3. How the authors could explain that situation ? Is there an increased lysosome number ?
Response 10:
We suggest that the lysosome numbers were not altered because the immunohistochemical (IHC) staining of LAMP1 in mature osteoclasts (after 3 days of RANKL stimulation) showed a similar fluorescent intensity (the data not shown) between the GFP- and GFP-Rab11A-expressing osteoclasts. Therefore, we conjecture that the increased expression levels of Cathepsins B and D were due to their increased accumulation in lysosomes, rather than due to the increased number of lysosomes in osteoclasts.
- Line 414, densitometry is mentioned although such information are not given in the figure 7.
Response 11:
We are sorry for our mistyping. We deleted the sentences “densitometry….”.
- Why, line 492 and following, the concluding remarks concern the macrophages ? It only was question of osteoclast throughout this manuscript.
Response 12:
We are sorry for our mistyping. We would like to replace the word “macrophage” with the word “osteoclasts”, based on the reviewer’s comment. The word revision was done in P17L503.
- Could the authors extrapolate on the way Rab11A would be expression in macrophages as they differenciate from a commun precursor in a kind of competition with osteoclasts?
Response 13:
We would like to briefly describe our hypothesis to elucidate the physiological role of Rab11A for osteoclast differentiation regulation as follows:
In the pre-osteoclasts, RAW-D cells, and/or bone marrow macrophages, Rab11A expression level is quite low. Upon RANKL stimulation, while c-fos and NFATc-1 would be transiently increased at an early stage of osteoclast differentiation, Rab11A was strongly increased at a late stage. It has been well-known that these transcription factors, NFATc-1, and c-fos, are transiently translocated into nuclei to promote the transcription process of the specific genes including trap, ctsk, and mmp9. At a late stage of osteoclast differentiation (after 3 days of RANKL stimulation), TRAP, CTSK, and MMP9 would be secreted into the bone milieu for bone resorption. We always believe that osteoclast differentiation would be dynamically regulated to ensure that the bone remodeling process would be maintained stably. More particularly, once mature osteoclasts are formed, there are a variety of inhibitory mechanisms underlying osteoclast differentiation. Here, we showed that Rab11A levels were reached maximum once the mature osteoclasts were formed. At the late stage, Rab11A would be mediating the surface down-regulation of c-fms and RANK receptors and dictating the vesicular transport of these receptors to lysosomes for degradation. Once c-fms and RANK surface receptors were abolished, osteoclast differentiation would be weakened, causally alleviating bone resorption. Therefore, in our study, we not only found that Rab11A was a negative regulator of osteoclast differentiation, more importantly, we elucidated the Rab11A-mediated transport route of c-fms and RANK surface receptors to lysosomes via the axis of early endosomes-late endosomes-lysosomes, subsequently declining osteoclastogenesis, eventually stabilizing bone resorption phase.
Reviewer 4 Report
Authors have identified a novel role of the Rab11A as a negative regulator of osteoclastogenesis.
Some minor comments on the article
- Using si RNA to knockdown Rab11A is an important method, it would be good if the authors can present the data on the M-CSF and RANKL osteoclasts control without any control siRNA treatment to compare the Rab11A inhibition induced osteoclastogenesis.
- c-fms and RANK receptor are downregulated when the osteoclasts are formed and osteoclasts are less stable and die after formation. Could Rab11A be linked to apoptosis in osteoclasts
- Authors can explain more clearly in conclusion on how Rab11A related mechanism help in developing future therapies.
Author Response
Authors have identified a novel role of the Rab11A as a negative regulator of osteoclastogenesis.
Some minor comments on the article
We appreciate this reviewer for the peer review and are delighted to response point-by-point below
- Using si RNA to knockdown Rab11A is an important method, it would be good if the authors can present the data on the M-CSF and RANKL osteoclasts control without any control siRNA treatment to compare the Rab11A inhibition induced osteoclastogenesis.
Response 1:
We thank the reviewer’s comment. However, we did not have the data showing the mock (not control siRNA)-treated osteoclasts to compare with Rab11A silencing-induced osteoclastogenesis.
- c-fms and RANK receptor are downregulated when the osteoclasts are formed and osteoclasts are less stable and die after formation. Could Rab11A be linked to apoptosis in osteoclasts.
Response 2:
We strongly appreciate the reviewer’s comment. It is a very important point. As we demonstrated that Rab11A overexpression down-regulated the surface levels of c-fms and RANK receptors in osteoclasts, feasibly causing an apoptotic consequence of osteoclasts. Therefore, it would be one of our future research goals in order to fully elucidate the physiological role of Rab11A for osteoclast differentiation regulation.
- Authors can explain more clearly in conclusion on how Rab11A related mechanism help in developing future therapies.
Response 3:
As we obviously explained, Rab11A overexpression would be negatively regulating osteoclast differentiation, thereby weakening bone-resorbing activity. The most well-characterized pathological bone disease, osteoporosis, is caused by perturbation of osteoclast activity, commonly gain of the function thereof; therefore, increased bone resorption, ultimately triggering a net loss of bone. Therefore, it would be necessary to develop potent drugs to promote Rab11A expression in osteoclasts, which possibly enables to alleviate the severe effects caused by osteoporosis disease.